# An Enhanced Multiplication of RF Energy Harvesting Efficiency Using Relay Resonator for Food Monitoring

**DOI:** 10.3390/s19091963

**Published:** 2019-04-26

**Authors:** Xuan-Tu Cao, Wan-Young Chung

**Affiliations:** Department of Electronic Engineering, Pukyong National University, Busan 608-737, Korea; tuxuancao@pukyong.ac.kr

**Keywords:** smart sensor tag, equivalent carbon dioxide sensor, battery-less sensor tag, meat freshness monitoring, relay resonator

## Abstract

Recently, radio frequency (RF) energy harvesting (RFEH) has become a promising technology for a battery-less sensor module. The ambient RF radiation from the available sources is captured by receiver antennas and converted to electrical energy, which is used to supply smart sensor modules. In this paper, an enhanced method to improve the efficiency of the RFEH system using strongly coupled electromagnetic resonance technology was proposed. A relay resonator was added between the reader and tag antennas to improve the wireless power transmission efficiency to the sensor module. The design of the relay resonator was based on the resonant technique and near-field magnetic coupling concept to improve the communication distance and the power supply for a sensor module. It was designed such that the self-resonant frequencies of the reader antenna, tag antenna, and the relay resonator are synchronous at the HF frequency (13.56MHz). The proposed method was analyzed using Thevenin equivalent circuit, simulated and experimental validated to evaluate its performance. The experimental results showed that the proposed harvesting method is able to generate a great higher power up to 10 times than that provided by conventional harvesting methods without a relay resonator. Moreover, as an empirical feasibility test of the proposed RF energy harvesting device, a smart sensor module which is placed inside a meat box was developed. It was utilized to collect vital data, including temperature, relative humidity and gas concentration, to monitor the freshness of meat. Overall, by exploiting relay resonator, the proposed smart sensor tag could continuously monitor meat freshness without any batteries at the innovative maximum distance of approximately 50 cm.

## 1. Introduction

With the frequent occurrence of food safety incidents, consumers are paying attention to food safety. They generally prefer food products that are safe, fresh, and of high quality. Therefore, the safety of food products is a critical issue for customers when making purchases. Food safety has usually been evaluated by relying on the senses with regard to variations in appearance, texture, smell, and color. However, these methods are inaccurate to evaluate the freshness, potential harmfulness, and meat shelf-life of food products. Food freshness traceability systems are required to improve food safety guarantees, quality levels, and product integrity along the food supply chain.

With the development of advanced technology trends, many methods to monitor food freshness in a quantitative manner have been proposed. One promising method of food monitoring was to use a single sensor module such as H_2_S gas sensor, CO_2_ gas sensor, relative humidity and temperature sensor, and color sensor [1,2,3,4,5,6]. In these proposed sensor devices, energy which powered the sensor was not the major consideration. Accordingly, these devices were operated in a wired or wireless manner and were powered by batteries. However, in such devices, batteries had several disadvantages: they must be either replaced or recharged periodically from fixed power sources, and they have relatively bulky sizes and weights. Although a sensor was low cost, compact in size, and consumes a small amount of power, these smart sensor devices still could not operate continuously without using batteries. Moreover, the transmission distance was also limited by power issues.

To overcome the limitations of batteries, techniques include harvesting ambient energy to either recharge the battery or even to directly power specific electrical loads using solar, kinetic, radio frequency, or thermal energy harvesting [7,8]. By using these methods, energy can be harvested via wireless energy transfer, where energy is transferred from the source to the transponder to supply power for transponder operation. The radio frequency energy harvesting (RFEH) technique, is known as a non-contact power transmission, has a sustainable power supply from a radio environment, and is a promising solution to power energy-constrained wireless networks. 

The RFEH system uses magnetic resonance coupling wireless power transfer (WPT), in which the antennas have the same resonant coupling based on the resonance circuits and near-field magnetic coupling concepts [9,10], for wireless energy transfer. Therefore, by using non-radiative WPT for wireless energy transfer, the magnetic fields, which capture the transmitted magnetic fields put through rectification, can create DC power for electrical applications [11]. The system can wirelessly charge the receiver with a distance transmission up to 10 times the antenna diameter [12]. This can be applied to wireless charging systems such as RFID applications [9,10,11,12], household electronics [13], medical implants [14], and electric vehicles [15] for high efficiency, non-radiative, and mid-range energy transfer.

There are two major passive RFID system types, high frequency (HF) and ultra-high frequency (UHF). The performance of a RFID system is strongly dependent on the environment between the reader and the tag where the communication occurs. Unlike HF, UHF can offer both near-field and far-field read ranges. However, owning to some HF technology’s characteristics such as near-field inductive coupling and homogeneous shape of field distribution make it less susceptible to the environmental noise and electrical interference. In contrast, UHF’s far-field technology does radiate real power via electric field, which makes it vulnerable to interference from the surrounding environment (i.e., metal, liquid). Moreover, [16] suggests that HF has more advantages over UHF at the item level. According to it, the HF technology is highly recommended to achieving item-level identification tracking for pharmaceutical and healthcare applications. Hence, in the current work, we select HF technology for our proposed food monitoring system owning to its above-mentioned advantages over the UHF technology.

Most of above-mentioned RFEH systems cannot avoid the limits of the reading range because of the harvested power, and the performance gradually decreases with increasing distance between the transmitter and receiver [17,18]. Therefore, many researchers have attempted to improve the power transfer efficiency. Recently, in order to achieve a good compromise between efficiency and transfer distance, some studies pointed out that the use of a relay resonator [19,20,21,22] based on the maximum energy efficiency principle should be considered. In these WPT systems, when the relay resonator is employed between the transmitter and receiver resonators, the relay resonator generally helps to focus the magnetic flux more effectively and efficiently over a relatively long distance, thereby enhancing the overall power transfer efficiency as well as extending the reading range [21], [22,23]. 

When we apply this wireless power transfer system using a relay resonator (WPT-R) to an energy harvester (RFEH-R system), we can harvest more power than the typical energy harvesting system. For this purpose, a comparison of WPT-R theory is analyzed to examine that the proposed relay-based system gets higher power efficiency than with a conventional energy harvesting system. The placement of the relay resonator should be adjusted depending on the quality factor of the Tx, Rx, and relay resonator, and the distance between the transmitter and receiver, to optimize the performance of the entire system.

The resonance frequency of the resonators is applied at a fixed operating frequency of 13.56 MHz. Then, the theory is designed via experiments using the relay resonator obtained from practical prototypes. Through the experiments, we demonstrate that the measured power achieved by the RFEH-R system is higher than that of the conventional energy harvesting system. 

For food freshness monitoring, the development of a real-time food freshness monitoring system is proposed using the proposed relay-based energy harvesting system. In order to monitor meat freshness, temperature, humidity, and gas concentration have been widely used [24,25,26,27,28]. We carefully chose the sensor to optimize the power consumption and the size of the entire circuit. Sealed environments are applied to maintain the gas proportion. Thus, a smart sensor system that consists of an RFEH-R module and a Sensing and Communication module is attached inside sealed meat packages for meat freshness monitoring.

The harvested energy is used to power the smart sensor load, whose function is to collect, process, and transmit the sensing data to the reader. When using the proposed energy harvesting system, the proposed smart sensor tag can supply power to the entire circuit without using any batteries. The results of the gas measurement are visually transmitted to the reader, which uses this data to evaluate the food freshness level of the meat. By employing these sensing data, the proposed battery-less food freshness system can predict the freshness level of meat products. 

The contents of the paper are organized as follows. In Section 2, we analyze the highly efficient and harvested power of the proposed system using a relay resonator, and compare it to a typical system by using an equivalent circuit model. In Section 3, the architecture of the overall proposed battery-less smart sensor system is presented. In Section 4, experimental results are examined to evaluate the highly efficiency of the proposed system and the sensing data that are used to evaluate food freshness in environments of various temperatures. Finally, we conclude the paper in Section 5.

## 2. Materials and Methods

To develop a battery-less smart sensor system, near-field wireless power transfer (WPT) technology is a promising technology that can power up the sensor load through the electromagnetic field effect without connecting to the main supply power or using a battery. Near-field WPT technology can be classified into three types: inductive coupling, magnetic resonant coupling, and capacitive coupling. These rely on the resonant frequency, efficiency, and transfer distance [29]. Among these, inductive and capacitive coupling mechanisms exhibit higher efficiency than magnetic resonant coupling mechanisms.

However, these methods have limited power transfer distance. Magnetic resonant coupling mechanisms typically work in high-frequency bands and can be achieve high system performance and a larger power transfer distance when the transmitter and receiver resonator resonate at the same resonance frequency. Therefore, magnetic resonant coupling wireless power transfer systems are chosen for this study. In this section, two WPT systems with and without a relay resonator are analyzed in detail to consider the high transmission efficiency and larger power of the WPT-R system.

### 2.1. The Wireless Power Transfer System Without Relay Resonator

Figure 1 shows a traditional RF power transfer system used for low-power applications. The illustration model of magnetic energy direction and density is shown in Figure 1a. It consists of a transmitter and a receiver side to provide the energy to supply the load. In the following notations, the symbols Tx and Rx refer to the indices for the transmitter and receiver, respectively.

A detailed analysis of magnetically coupled resonators in a traditional WPT system without a relay resonator, an equivalent circuit model is shown in Figure 1b. It provides a convenient reference for analysis of the transfer characteristics of a magnetically coupled resonator system. The Tx is powered by an alternative voltage source vs. and a source resistor R_S_, while the Rx is connected to a load resistor R_L_. Moreover, L_i_, and r_i_ are the self-inductance, self-capacitance, and internal resistance of the i^th^ antenna. A capacitor C_i_ is added make each resonator resonate at the desired frequency of 13.56 MHz (where i = 1, 2) [30]. The angular resonance can be expressed as
(1)$$\omega_{0}=2\pi f_{0}=\frac{1}{\sqrt{L_{1}C_{1}}}=\frac{1}{\sqrt{L_{2}C_{2}}}$$

The distributions of magnetic flux in a traditional WPT system are shown in Figure 1a. In this case, the magnetic flux, which is generated by I_1_ from the transmitter resonator, goes through the receiver resonator and then induces the alternating current I_2_. The magnetic flux between the two resonators can be represented by a mutual inductance [22]:(2)$$M_{12}{=k}_{12}\sqrt{L_{1}L_{2}}≅\frac{{\pi \mu N}_{1}N_{2}r_{1}^{2}r_{2}^{2}}{{2(d}_{12}{+r}_{2}{)}^{\frac{3}{2}}}$$
where k is a coupling coefficient. In Figure 1b, by applying Kirchhoff’s voltage law (KVL) and current laws, the relationship between the currents through each antenna, and the voltage applied to the power antenna, can be captured as the following equations:(3)$$\begin{array}{l} Z_{1}I_{1}+j\omega_{0}MI_{2}=V_{S}\\ j\omega_{0}MI_{1}+Z_{2}I_{2}=0 \end{array}$$
where $Z_{1}=R_{S}+r_{1}+jX_{1}$ is the resistance of the transmitter, $Z_{2}=R_{L}+r_{2}+jX_{2}$ is the resistance of the receiver. Xi is the reactance of the i-th resonator (where i = 1, 2).

However, in order to maximize the power transfer capability of the system, both the transmitter and the receiver resonator must resonate at the same resonant frequency, i.e., X_1_ = X_2_ = 0 [31]. Therefore, the current in each antenna can be calculated as
(4)$$\begin{array}{l} I_{1}=\frac{V_{S}Z_{2}}{\omega_{0}{}^{2}M^{2}+Z_{1}Z_{2}}\\ I_{2}=\frac{-jV_{S}\omega_{0}M_{12}}{\omega_{0}{}^{2}M^{2}+Z_{1}Z_{2}} \end{array}$$

An entire typical WPT system can be considered as a two-port network: an input fed by the source as the first port, and a second port that feeds the load and is output fed. Thus, the transferred power can be represented in terms of the linear magnitude S_21_. We can calculate the equivalent S_21_ scattering parameter as follows [32,33]:(5)$$S_{21}=2\frac{V_{L}}{V_{S}}\sqrt{\frac{R_{S}}{R_{L}}}=2\frac{-j\omega_{0}M\sqrt{R_{S}R_{L}}}{\omega_{0}^{2}M^{2}+Z_{1}Z_{2}}$$

V_L_ is the voltage across to the load resistor R_L_, given by V_L_ = R_L_I_2_. Therefore, the power transfer efficiency of a typical WPT system (the power delivered to the load provided by the available power from the source) can be expressed as in (6) [32,33]:(6)$$\eta ={\left|S_{21}\right|}^{2}=4\frac{R_{S}R_{L}}{Z_{1}Z_{2}}\frac{M_{12}^{2}A_{1}A_{2}}{{\left(1+M_{12}{}^{2}A_{1}A_{2}\right)}^{2}}$$
where the notation Ai represent for the ratio of Q_i_/L_i_. The quality factor Q_i_ of a resonator is represented as $Q_{i}=(1/Z_{i})\sqrt{L_{i}/C_{i}}$. By Equation (6), the power transfer efficiency can be expressed as a function of the quality factors, the operating frequency, the load, the quality factors, and mutual inductances. Assume that the resonators resonate at the same frequency, which is identical to the desired operating frequency of our wireless power transfer system. The parameter of each resonator and the load are also fixed. Therefore, the quality factors are considered as static parameters. In order to maximize the power transmission efficiency for extended power transfer, the mutual inductance becomes an important factor in achieving high harvested power as well as power transmission efficiency.

In a straight chain resonator system, the mutual inductance between two resonators is inversely proportional to the transmission distance, as shown in Equation (2). This means that if the distance between the transmitter and the receiver increases, the mutual inductance between two resonators will decrease correspondingly, and thus the transmission efficiency of the system will decrease. When the transmission distance is long enough, we cannot harvest the energy because the magnetic interaction of the magnetic fields between the two antennas goes quickly toward zero.

Increasing the quality factor of the resonators is another useful way to achieve high power transfer efficiency. An augmentation of the resonator parameters is applied. However, with the fixed diameter in a practical system, the quality of each resonator is not changed. Therefore, the desired system, which uses a typical WPT system without a relay resonator, is not suitable for high-efficiency power transfer and long transmission distance.

### 2.2. The Proposed Wireless Power Transfer System With A Relay Resonator

In order to increase the efficiency as well as extend the reading range of the system in order to increase the energy efficiency and transfer distance, we considered a relay-resonator-based magnetic resonance wireless power transfer (WPT-R) system. The WPT-R system can possibly overcome the drawbacks of the conventional WPT system, as the transmitter and receiver are far away from each other. A model of the proposed WPT-R system and modeled equivalent circuit are shown in Figure 2. The schematic is composed of three resonant circuits corresponding to the three resonators: a transmitter (Tx) resonator, receiver (Rx) resonator, and a relay resonator. The resonators are linked magnetically, which is characterized by mutual inductances M_12_, M_23_, and M_13_. This section demonstrates that under specific design conditions, the high-energy efficiency of the WPT-R system under a specific design condition could be higher than that of the traditional system.

To fairly compare the efficiency of the two systems, we use identical factors: resonator parameters in the system without using a relay resonator, distance range, and measuring environment. A relay resonator is added to the center axis line between the Tx and Rx resonators. The relay resonator restricts the dissipation of the magnetic flux and generates an enhanced amount of magnetic flux from the Tx to the Rx. It consists of a rectangular loop antenna connected to a capacitor that allows the relay resonator to resonate at a fixed resonating frequency (13.56 MHz).

At a long distance range between Tx and Rx, assume that the cross coupling between the Tx and Rx resonators can be neglected [34]. Then, we can again set up the following equations from Kirchoff’s voltage law for the WPT-R system:(7)$$\begin{array}{l} Z_{1}I_{1}+j\omega_{0}M_{12}I_{2}+j\omega_{0}M_{13}I_{3}=V_{S},\\ j\omega_{0}M_{12}I_{1}+Z_{2}I_{2}+j\omega_{0}M_{23}I_{3}=0,\\ j\omega M_{13}I_{1}+j\omega M_{23}I_{2}+Z_{3}I_{3}=0. \end{array}$$

By solving the full KVL equations on the equivalent circuit in Figure 2b, the receiver current at the resonant frequency is found to be
(8)$$\begin{array}{l} I_{1}=\frac{V_{S}/Z_{1}}{1+\frac{M_{12}^{2}A_{1}A_{2}}{1+M_{23}^{2}A_{2}A_{3}}}\\ I_{2}=\frac{jV_{S}\omega_{0}/\left(Z_{1}Z_{2}\right)}{\frac{1}{M_{12}}+\frac{M_{23}^{2}}{M_{12}}A_{2}A_{3}+M_{12}A_{1}A_{2}}\\ I_{3}=\frac{V_{S}\omega_{0}^{2}/\left(Z_{1}Z_{3}\right)}{\frac{1}{M_{12}M_{23}Q_{2}}+\frac{M_{12}}{M_{23}}A_{1}+\frac{M_{23}}{M_{12}}A_{3}} \end{array}$$
where A_i_ = Q_i/_L_i_ for i = 1, 2, 3. Similarly as the typical WPT system, the equivalent S_31_ scattering parameter can be calculated as
(9)$$S_{31}=2\frac{V_{L}}{V_{S}}\sqrt{\frac{R_{S}}{R_{L}}}=2\frac{\omega_{0}^{2}\sqrt{R_{S}R_{L}}/Z_{1}Z_{2}Z_{3}}{\frac{1}{M_{12}M_{23}}+\frac{M_{23}}{M_{12}}A_{2}A_{3}+\frac{M_{12}}{M_{23}}A_{1}A_{2}}$$

Therefore, the power transfer efficiency which is expressed in terms of the S_31_ scattering parameter, can be obtained as follows
(10)$$\eta ={\left|S_{31}\right|}^{2}=\frac{R_{S}R_{L}}{Z_{2}{}^{2}}{\left(\frac{A_{1}A_{3}}{\frac{1}{M_{12}M_{23}}+\frac{M_{23}}{M_{12}}A_{2}A_{3}+\frac{M_{12}}{M_{23}}A_{1}A_{2}}\right)}^{2}$$

When resonator parameters are based on the design of an identical system, the self-inductance and quality factors of each resonator are also not changed. Therefore, we do not need to consider the value of Ai. In this case, the transferred power is higher if the resonant frequency is synchronously at the desired resonant frequency of 13.56 MHz. In order to maximize the harvested power efficiency, the optimum position of the relay resonator can be found in (10). Assume that our identical design for the relay resonator is greatest and that the quality factor is high. Therefore, based on Equation (10), the harvested power efficiency is maximized when
(11)$$\frac{M_{23}}{M_{12}}=\frac{A_{1}}{A_{3}}$$

With regard to Equation (11), the mutual inductances between the resonators are proportional to the quality factor of the transmitter and receiver resonators in order to maximize the harvested power efficiency. Otherwise, the quality factor of the transmitter resonator is better than the quality factor of the receiver resonator which designed by the identical system. Therefore, the relay resonator should be placed near the receiver resonator. To demonstrate the higher harvesting power of the WPT-R system and the typical WPT system, a power verification is shown in the next section.

### 2.3. Power Verification: Comparison Between Simulation and Experiment

In order to verify the effect of the relay resonator on a WPT system, the power efficiency value of the entire system can be used for comparison. Therefore, computer simulations are carried out to compare the energy efficiency of the two WPT systems using the equivalent circuits in Figure 1b and Figure 2b. For practical application, we consider a circuit in which the resonator parameters are similar to those in our proposed practical system. The simulations were performed using Agilent Advanced Design software (ADS), and the simulation parameters for each resonator are specified in Table 1.

The Tx resonator are the big antenna, which is tuned by 19 copper turns and is connected to several fixed-value capacitors and a tunable capacitor. The Rx resonator and relay resonator are tuned by seven copper turns that are fabricated on an FR4 substrate. Both resonators were carefully tuned to achieve the desired resonant frequency. With the self-capacitance of the spiral loop, the measured self-resonant frequency was 13.56 MHz. All antennas were aligned in the coaxial direction, and the distance between the transmitter and receiver resonators varied from 5 to 80 cm in 5-cm steps. However, when the reading range varied, there was a corresponding change in M. With increasing distance, a misalignment of Q and M could occur with any different relative position. In practice, we change M by varying the distance between transmitter, receiver, and relay resonators.

The WPT and WPT-R system were implemented by conducting an investigation on the effect of the relay resonator position that influenced the harvested power efficiency. Modeling and numerical analysis were carried out using Ansys HFSS and Matlab software by changing d from 5 to 80 cm. And the position of relay resonator, which presents by notation d1, is the distance from the Tx and relay resonator.

The optimum position of a relay resonator between Tx and Rx can be found by observing the power efficiencies at the various placements by calculating the Equations (6) and (10). Figure 3 and Figure 4 show the simulation result using MATLAB when the distance d_1_, the distance from the Tx and relay resonator, is changing versus to the distance d between the Tx and Rx. It is observed that the power efficiency is decreasing where d is from 10 to 80 cm. At the d fixed to 50 cm, the optimum relay position is found at 46.8 cm from the Tx with the efficiency of 10.2% from the simulation result. When Rx is in close distance at d = 5 cm from Tx, there is a frequency splitting effect at the resonant frequency [35]. Hence, the measured efficiency at d = 5 cm is lower than the measured efficiency at d = 10 cm.

Furthermore, the simulation and measurement of harvested power are considered to reinforce the high performance of the proposed WPT-R system. For a harvested power simulation, we again use the Agilent ADS software to estimate the harvested power. In order to estimate the greatest possible power obtained, we created a position loop for the relay resonator with varying distances between the Tx and Rx resonators. Based on this method, we can use the estimated position for power experiments in practical applications.

The experiments are measured by a power experiment, which is implemented in the range d from 5 to 80 cm by pulling the Rx resonator from the Tx resonator with the optimized position d_2_ between relay resonators, as shown in Figure 5. All positions of the resonators are optimized by using the estimated position that can obtain the maximum power in the simulated analysis. For accurate measurements of the transfer efficiency at different distances, we use a Vector Network Analyzer (Agilent E5062A) to obtain the power value.

Figure 5 shows the harvested power as plotted against the distance between the Tx resonator and Rx resonator. The lines indicate the simulated results, while the markers represent the experimental results. As shown in the figure, the harvested power of both systems dramatically decreases as d decreases. However, the harvested power of the proposed WPT-R system is greater, even in the long range of d, compared to the typical WPT system. For example, the obtained power of the WPT-R system at 50 cm is around 35 µW higher than the 4.2 µW of the typical system. It can be clearly seen that the proposed WPT-R system can ensure a greater harvested power than the typical system.

By using a relay resonator in a WPT system, the proposed WPT-R system allows for significant improvements in both system efficiency and harvested power. Therefore, we can implement this method along with an RF energy harvesting circuit for a wide range of applications.

## 3. Food Freshness System Design and Implementation

In order to monitor the food freshness level, a real-time full passive food freshness monitoring system has been presented. Figure 6 shows a block diagram of the proposed passive smart sensor system, which includes a 4-W reader to communicate as well as supply power to the smart sensor tag using an RF signal. The relay resonator relays the RF signal strength, thus improving the energy supply for the smart sensor tag.

Our passive smart sensor module, operating in an HF band of 13.56 MHz, is composed of two Sections: (1) RF energy harvesting using a relay resonator and voltage management section, and (2) a digital sensing and signal transmitting section. The proposed passive smart sensor module was utilized to monitor meat freshness at a distance of 50 cm.

Meat spoilage depends mostly on temperature, which to a large extent controls the bacterial and autolytic breakdown. On the other hand, the equivalent CO_2_ (eCO_2_) gases, which is released during the decay of food, were also utilized to assessment the meat freshness. In order to evaluate the meat freshness, a temperature, humidity, and eCO_2_ gas sensor was used to capture the changing environment inside the sealed meat containers. After that, all sensing data were sent to the microcontroller unit (MCU) for processing and IC tagging for transmission. Finally, all sensing data were transmitted wirelessly to the reader using RF waves. However, the fact that there is no battery to supply power to the entire circuit is the main issue in this study. Therefore, we have to choose components that have the smallest energy consumption.

To overcome this issue, we need to carefully choose the components to ensure the circuit’s operation does not use any batteries. A digital humidity sensor that integrates humidity and temperature sensors and provides excellent measurement accuracy at very low power was used. This sensor obtains the relative humidity and temperature inside the meat storage location. To detect the eCO_2_ gas concentration, an ultralow-power digital gas sensor solution that integrates a metal oxide (MOX) gas sensor was used. The eCO_2_ sensor includes an analog-to-digital converter (ADC) and an I^2^C interface to communicate with the MCU. The sensor supports intelligent algorithms and multiple measurement mode for sensor measurements that are optimized for average low-power consumption at around 1 mW during an experiment cycle.

Both sensors support an I^2^C interface to communicate with the MCU. The low-power consumption IC tag, which complies with the ISO 15,693 standard for wireless communication with the reader at a 13.56 MHz frequency, operates well in full passive mode. An extremely low-power microcontroller was selected to process data from the sensors and to write the data to the memory of the IC tags via the I^2^C interface. The total calculated power consumption of the abovementioned components (including a temperature and humidity sensor, eCO_2_ gas sensor, voltage regulator, microcontroller, and IC tag chip) is around 1.5 mW which are listed in Table 2.

As analyzed in Section 2, the proposed energy transfer methodology ensures that the three-resonator system is more energy efficient than a conventional RF wireless power transfer system. However, the RF energy, which is harvested from the reader, is not enough to the operation of the entire sensing system. Therefore, an RF energy harvesting system using a relay resonator (RFEH-R system) is designed. Figure 7 shows a block diagram of step-up converter and power management of the proposed RFEH-R system.

The RFEH-R system consists of a relay resonator, a rectangular antenna, a voltage regulator, energy storage, and a voltage supervisor. The voltage regulator converts the harvested RF signals into a DC signal, and usually includes Schottky barrier diodes for a high performance of RF-to-DC rectification [36]. The relationship between the harvested AC voltage (V_in_) and the rectified voltage (V_rect_) is calculated based on [37]:(12)$$Vrect=n\times \left(2\times V_{in}-2\times V_{diode}\right)$$
where n = 3 is the number of stages of the multiplier. 

Most components of the smart sensor tag require a minimum DC voltage of 1.8 V to operate. In order to provide enough voltage to turn on the sensor module, a three-stage multiplier was used, particularly at long distances. Further, a 0.1-F super-capacitor was used to sufficiently store the DC power of the minimum voltage to operate the entire circuit. However, when the reading range is extended to 50 cm, the super-capacitor needs a long time for charging. The MAX6433 (Maxim, USA) voltage supervisory chip is used to control the charging time and the uptime. The voltage supervisory chip is configured to operate at two threshold voltages: the higher threshold voltage (VHT) and lower threshold voltage (VLT). The higher threshold voltage and lower threshold voltage are 2.0 V and 3.7 V, respectively. Using a switch to control common ground, the voltage supervisory chip can control the charging and discharging of the super-capacitor. In addition, it also ensures the impedance matching at RF energy harvester part. If the voltage of the super-capacitor reaches the higher threshold voltage, it will supply power to the entire circuit. On the other hand, if the voltage of the super-capacitor is below the lower threshold voltage, it will interrupt to supply power and continuous to charging the super-capacitor. A photograph of the battery-less smart sensor tag is shown in Figure 8.

Two environments were set up to measure the food freshness level at cold chamber and room temperatures of 4 °C and 22 °C, respectively. These environments were used to monitor the relationship between pork freshness/spoilage and the storage environment during the decay process. Samples of pork were purchased from a standard market at high quality and were immediately prepared for the experiment. Ten samples of pork were placed in sealed packages to maintain the gas proportion. A total of 100 g of each was selected to determine the sensing signal and freshness level of the pork.

In a sealed pork package, the decomposition of pork product progresses owing to microorganisms, and volatile organic compounds such as eCO_2_ gasses and TVB-N are produced. The eCO_2_ gases increase over time within the headspace of the sealed pork package. The concentration of emitted eCO_2_ gas is monitored by the sensor modules attached to the headspaces of the pork packages. These sealed package samples are separated into the two temperature environments (room temperature and refrigerator temperature) during experiments over eight days. A laboratory room and a refrigerator are prepared for storing the 10 sealed packages for measurement experiments.

Figure 9 shows the experimental setup for the proposed meat freshness monitoring using a relay resonator. The smart sensor tag, which is energized by the 4-W reader, is placed at a distance of d = 50 cm from the reader. The relay resonator is placed at the distance of d_2_ = 3.2 cm from the smart sensor tag, as calculated in Section 2. The centers of the three resonators are located on the same straight line to optimize the harvested power.

Research manuscripts reporting large datasets that are deposited in a publicly available database should specify where the data have been deposited and provide the relevant accession numbers. If the accession numbers have not yet been obtained at the time of submission, please state that they will be provided during review. They must be provided prior to publication.

## 4. Experimental Results and Discussion

The power measurement setup of the proposed passive smart sensor tag with/without a relay resonator is shown in Figure 10. We measured the harvested power using a Tektronic oscilloscope. The harvested RF signals through the transmitter antennas are indicated by blue lines, while the measured DC signals (which are converted and multiplied from the harvested RF signal) are represented by green lines. It is clear that in the proposed RFEH-R system, the total power transmission is approximately five times larger than that of the traditional power transfer system. This is because the relay resonator can help obtain a significant amount of RF signal, and hence the harvested DC power of the RFEH-R system is larger than that of the typical RFEH system.

The DC signals obtained by the RFEH-R system are accumulated in a super-capacitor until the loaded energy is enough to operate the sensor module. Figure 11 shows the operation of power management for the battery-less system. At 50 cm, the RFEH-R takes approximately 94 s for the 0.1-F super-capacitor to be charged until V_HT_ is reached. From the practical results, the designed proposed RFEH-R system can operate fully in ~7 s without needing a battery. This indicates that it is better to use the relay resonator for RF energy harvesting to improve system efficiency and to extend the range of transmission.

For pork package samples in a room-temperature environment, the production of volatile organic compounds (VOCs) owing to food spoilage in the sealed packages containing pork is monitored during storage. Among these VOCs, for the eCO_2_ concentration gas, temperature sensing signals are collected automatically at time intervals of 3 min to predict the freshness level of the individual pork products. The increasing linearity of the eCO_2_ concentration gas during the eight days of experiments is shown in Figure 12.

The measured temperature is also observed in Figure 12. The fluctuations in the graph can be explained by changes in the measuring environment. The graph illustrates the considerably different gas sensing from zero after only eight days of storage. The eCO_2_ concentration gas is produced on the first day after slaughter during the autolytic growth of microorganisms. After three days, a large amount of the eCO_2_ gas concentration is identified in the sealed pork packages, and has reached 6000 ppm. From Day 4 to Day 8, the color of pork changes from red to gray, and then dark green.

Figure 13 shows the detection of the eCO_2_ gas concentration and temperature sensing signal in a refrigerator environment after eight days when the meat is stored at 4 °C. The graph with the results of the sensing data analysis during the experiments is comparable to the results reported in room-temperature conditions. There is a slower increasing in emitted gas concentrations when a sealed pork package is placed in the refrigerator compare with the emitted gas concentrations when the sealed pork package was put in the room environment. During storage in the refrigerator, the spoilage phenomenon started on Day 5, when there was a change in the color of the pork samples. The explanation might be a temperature deviation between the storage of pork samples in the refrigerator and room-temperature environments. A decrease in the room temperature to 4 °C reduced the gas emissions by half in the sealed samples and slowed the decomposition process. The experiment results show that the RF sensing data can be read accurately even at low temperatures.

Based on the freshness monitoring experiments, the RFEH-R module is periodically provided harvested power for continuously monitoring changes in the freshness of individual pork package samples. Therefore, the decay in quality of pork can be monitored non-destructively by using this method. Different sensing signals are seen at different temperatures, and monitoring the eCO_2_ gas emissions can be used to predict the pork freshness level. Hence, this smart sensor system can be used as a wireless measurement to monitor the sensing data that is present in a sealed food package.

## 5. Conclusions

In this study, we investigated the effects of a relay resonator on the multiplication of the power transfer. The theory and design criteria for the proposed harvesting system have been investigated and presented. From these results, we found that the quality factor of resonators and the optimal position of the relay resonator have a strong effect on enhancing the RF harvested-energy performance and transmission distance. This finding was applied in an RF energy harvesting module using a relay resonator (RFEH-R) to power a full passive smart sensor tag for pork freshness monitoring. Results showed that the proposed power transfer system using the relay resonator obtained better performance than a traditional power transfer system without a relay resonator in terms of energy efficiency.

Our proposed RFEH-R system, consisting of the 4-W reader, sensing module, relay resonator, and energy harvesting module, demonstrated a higher efficiency in comparison with that of the traditional RFEH (i.e., without relay resonator). The designed smart sensor tag was operated in passive mode within a long distance of approximately 50 cm at a high frequency of 13.56 MHz. The 50-cm range for the passive smart sensor tag is significantly longer than the typical range of 10 to 15 cm. Furthermore, by using the proposed RFEH-R system, the architecture of a full passive smart sensor tag for pork freshness monitoring in sealed environments was presented. During the measurement, the temperature, humidity, and eCO_2_ gas concentrations are collected and processed by the MCU for monitoring and predicting the quality of the food.

In summary, our proposed food freshness system can be applied as a battery-free sensor tag which helps to reducing the food spoilage rate and improving customer satisfaction, as well as operational efficiency. For the future work, we aim to design a range-adaptive radio frequency energy harvesting system for further investigating the effect of load variation and position of relay resonator between transmitter and receiver on the system performance.

## Figures and Tables

**Figure 1 sensors-19-01963-f001:**
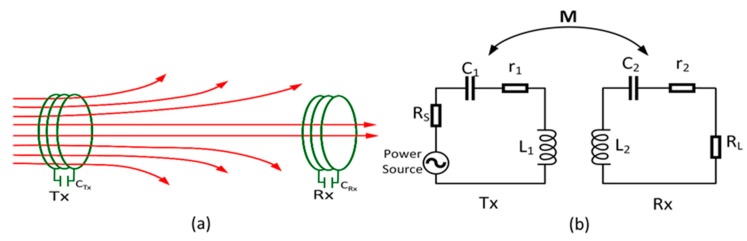
(**a**) The conventional radio frequency energy transfer system without relay resonator and (**b**) its equivalent circuit model.

**Figure 2 sensors-19-01963-f002:**
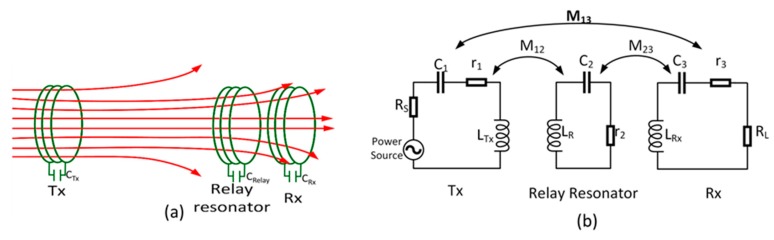
(**a**) The enhanced multiplication system with relay resonator and (**b**) its equivalent circuit model for energy harvesting system.

**Figure 3 sensors-19-01963-f003:**
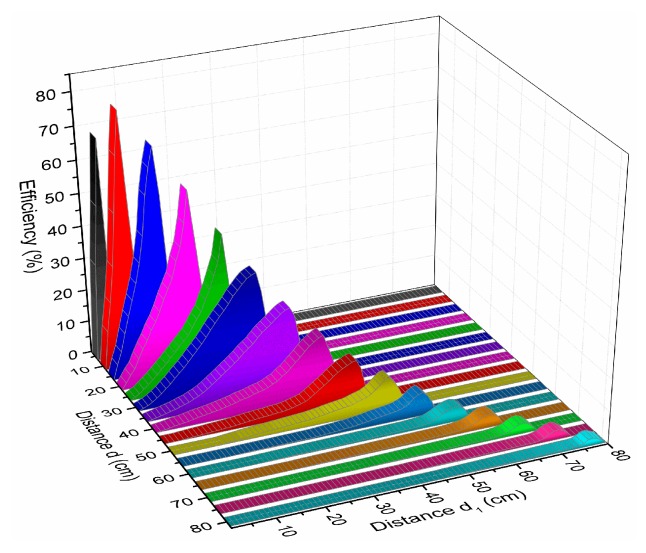
Power efficiency simulation results for the position of relay resonator from the transmitter with respect to the distance between transmitter and receiver resonators.

**Figure 4 sensors-19-01963-f004:**
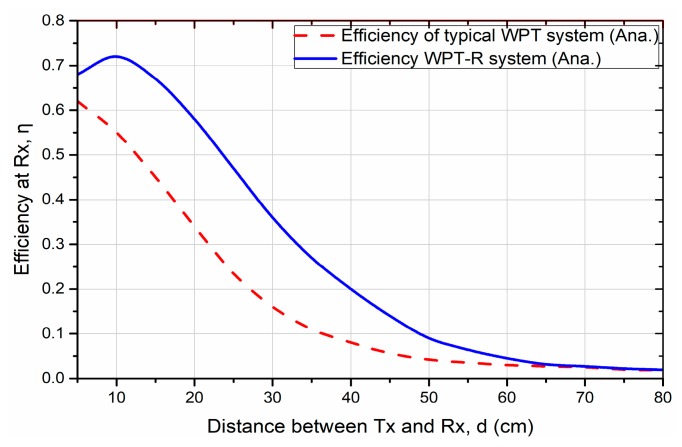
The comparison of simulated harvested power efficiency between typical wireless power transfer (WPT) and wireless power transfer using a relay resonator (WPT-R) systems.

**Figure 5 sensors-19-01963-f005:**
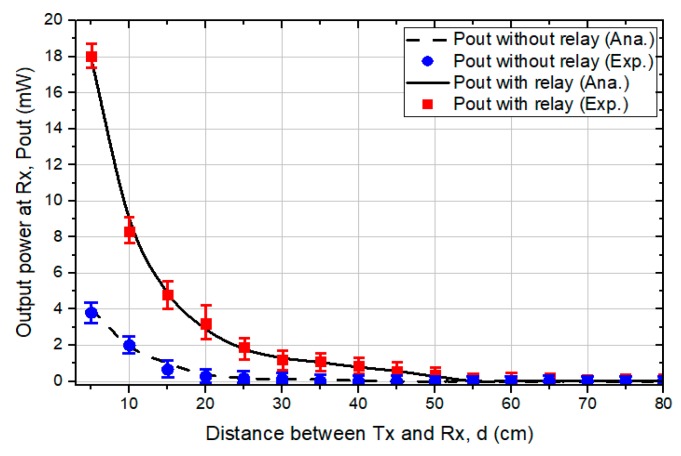
Comparison of simulated and experimental harvested power between typical wireless power transfer (WPT) and wireless power transfer using a relay resonator (WPT-R) systems.

**Figure 6 sensors-19-01963-f006:**
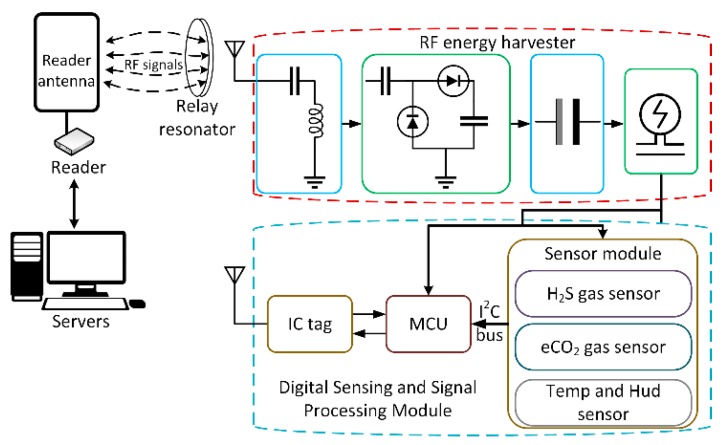
Block diagram of proposed passive smart sensor system.

**Figure 7 sensors-19-01963-f007:**
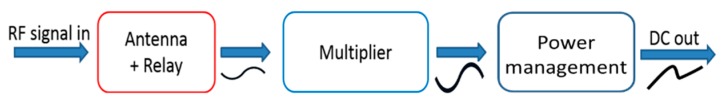
Block diagram of step-up converter and power manager for energy harvesting application.

**Figure 8 sensors-19-01963-f008:**
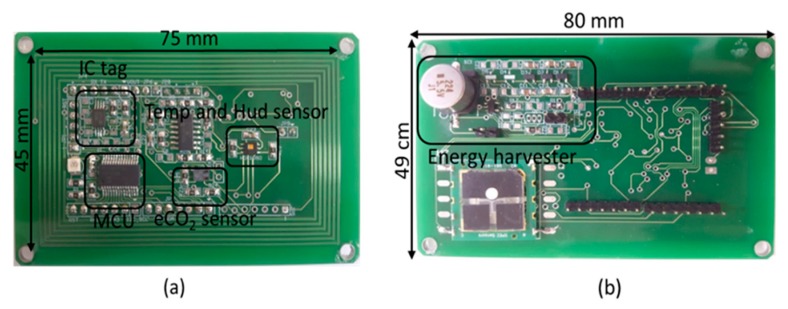
Photograph of wireless passive smart sensor module: (**a**) top layer and (**b**) bottom layer.

**Figure 9 sensors-19-01963-f009:**
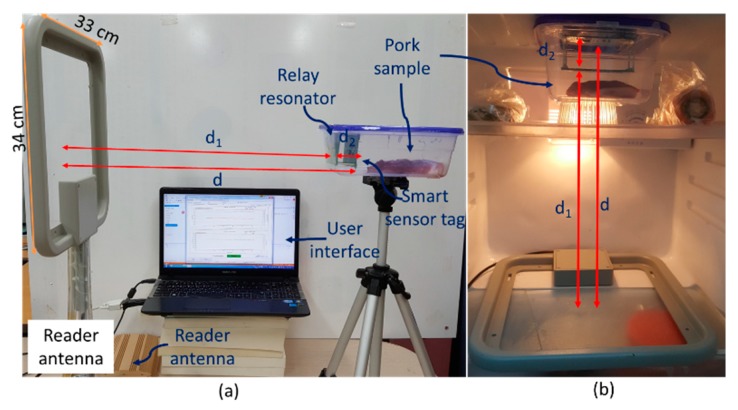
Experimental setup for proposed meat freshness monitoring using relay resonator at (**a**) room temperature and (**b**) refrigerator environments.

**Figure 10 sensors-19-01963-f010:**
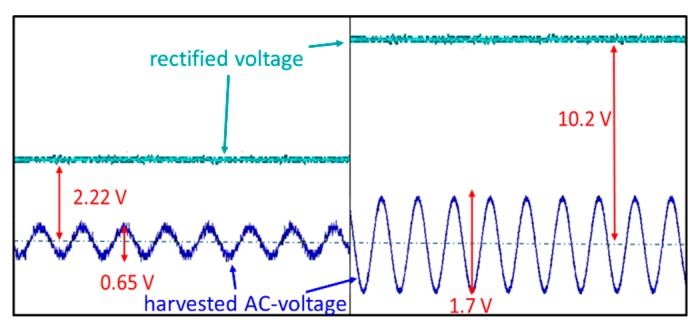
Harvested power comparison of two systems: (**a**) typical radio frequency energy harvesting system and (**b**) proposed radio frequency energy harvesting using a relay resonator system.

**Figure 11 sensors-19-01963-f011:**
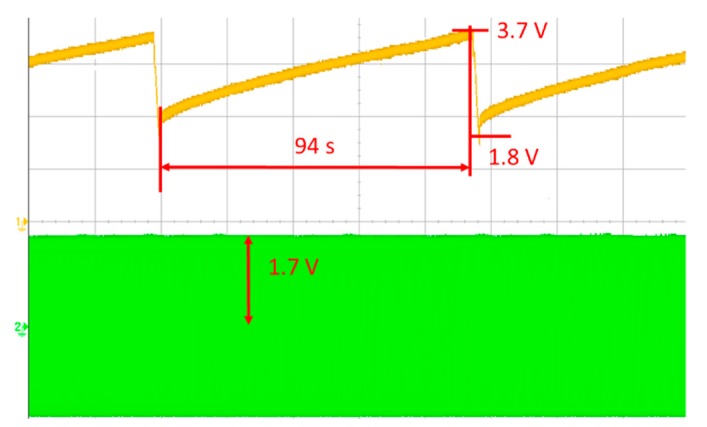
Operation of power management for battery-less system.

**Figure 12 sensors-19-01963-f012:**
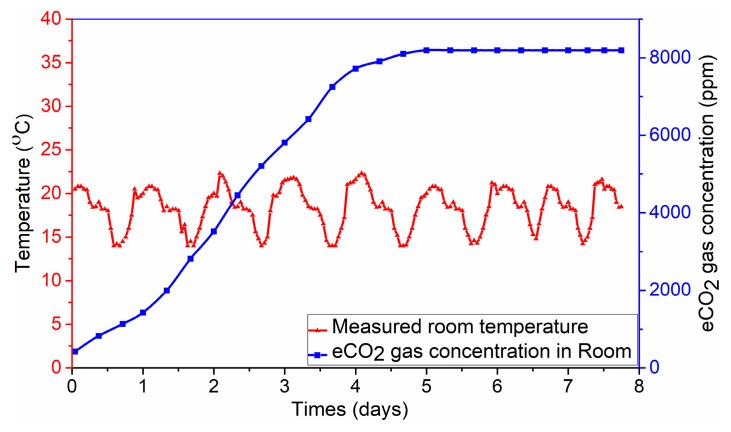
Food monitoring experimental results of measuring temperature and eCO_2_ gas concentrations in room environments.

**Figure 13 sensors-19-01963-f013:**
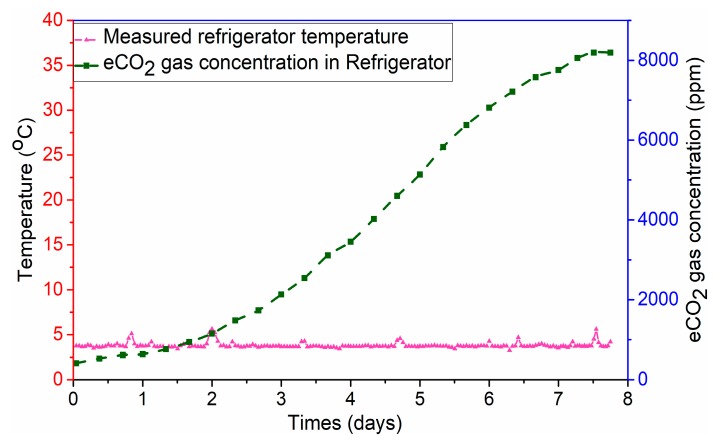
Food monitoring experimental results of measuring temperature and eCO_2_ gas concentrations refrigerator environments.

**Table 1 sensors-19-01963-t001:** Simulated electrical parameters of resonators.

Resonator	Transmitter	Relay	Receiver
N (turns)	19	6	6
Inductance (µH)	490	4.62	4.92
Resistance (Ω)	24.05	4.37	4.37
Resonant frequency (MHz)	13.56	13.56	13.56

**Table 2 sensors-19-01963-t002:** Average power consumption of the smart sensor tag operating.

Component	Component Type	Voltage Supply	Power Consumption
PIC16LF1513	Microcontroller	1.8–5.5 V	~90 µW
M24LR04E-R	IC Tag chip	1.8–5.5 V	~200 µW
SGP30	eCO_2_ sensor	1.62–1.98 V	~1 mW
HDC2010	Temperature and humidity sensor	1.62–3.6 V	~400 nW
MAX6433	Voltage supervisor	0.3–6 V	~1.8 µW
Total			~1.5 mW

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
