# Peer review of "An Enhanced Multiplication of RF Energy Harvesting Efficiency Using Relay Resonator for Food Monitoring"

_sensors, 2019, doi:10.3390/s19091963_

Round 1
Reviewer 1 Report
This paper describes a novel energy harvesting enhancement method using a relay resonator, and an implementation with a sensor combo circuit is demonstrated. The results agree well to the theoretical estimation and clearly shows the feasibility of the system. The reviewer agree to publish this paper except a very minor typo.
The section number of Conclusions should be 5.
Author Response
Reviewer#1, Concern # 1: The section number of Conclusions should be 5
Author response: We appreciate the reviewer for pointing out the mistake.
Author action: We updated the manuscript by changing the section number of Conclusion part:
“…sealed food package
5. Conclusions
In this study…”.
Reviewer 2 Report
The authors provide sound theoretical analysis and conducte comprehensive technical developments and experimental studies for the HF RFID relay to improve the energy harvesting efficiency and reading distance. It is very useful work which is of interest to people working on RF and IoT research and development. I think the authors need to clarify the folliwing questions: (1) Since UHF RFID can perform the same functiona and can conver even longer reading distance, what is the strength of this 15693-based standard based sensor approach ? (2) Since the presented work is going to be published, I think it is very helpful for the peer researchers to know the components used. Please give a comprehensive list of the major components used in the RFID sensor and the relay. (3) Please explain the relationship between harvested ac voltage and rectified voltage.Author Response
Reviewer#2, Concern # 1: Since UHF RFID can perform the same function and can cover even longer reading distance, what is the strength of this 13.56 MHz-based standard based sensor approach?
Author response: We appreciate the reviewer for this comment and have summarized the comparison between these two wavelength-based RFID systems in the following table for clarification:
HF RFID | UHF RFID |
Based on inductive coupling (using the magnetic field to transfer power and data): - Well-defined magnetic field - Smaller read zone but easier to control - Shorter maximum range but more reliable on an object made of metal | Based on capacitive coupling (using electric field): - Bigger read zone but harder to control and more complex because energy is sent over long distances - Vulnerable to interference from both metal and liquid - Better range, faster data rate but use more power and are less likely to pass through materials |
Longer wavelength: - High RF harvested power density - More able to penetrate in liquids due to less absorption - Suited for tagging liquid-bearing or carbon-composed products | Shorter wavelength: - Low RF harvested power density - Can be applied in water/liquid-bearing applications but its effective read range would be drastically reduced |
Near-field: - Reader emits magnetic field. When a transponder passes through, an electric current is created that powers the RFID tags and transmits data | Offer both near-field and far-field read ranges. However, for the near-field RFID: - Antennas generate magnetic field. Since the tag is closer to the antennas, near-field UHF has a narrower field of view and a shorter read range compared to the HF |
HF distribution fields are a homogeneous shape, which prevents communication gaps or blind spots, making it less susceptible to environmental influence | UHF field distribution tends to be in-homogeneous, resulting in blind spots, communication gaps, making the system more susceptible to environmental influence |
The air interface size and shape are independent of the surrounding environment | The size and shape of air interface is highly influenced by the surrounding environment |
Author action: The concern has been clarified and we also added the comparison of HF and UHF RFID systems in the introduction part:
“
… mid-range energy transfer.
There are two major passive RFID system types in high frequency (HF) and ultra-high frequency (UHF). The performance of a RFID system is strongly dependent on the environment between the reader and the tag where the communication occurs. Unlike HF, UHF can offer both near-field and far-field read ranges. However, owing to some HF technology characteristics such as near-field inductive coupling and homogeneous shape of field distribution it is less susceptible to the environmental noise and electrical interference. In contrast, UHF’s far-field technology does radiate real power via electric field make it vulnerable to interference from surrounding environment (i.e., metal, liquid). Moreover, [16] suggests that HF has more advantages over UHF at the item level. According to it, the HF technology is highly recommended to achieving item-level identification tracking for pharmaceutical and healthcare applications. Hence, in the current work, we select HF technology for our proposed food monitoring system owning to its above-mentioned advantages over the UHF technology.
Most of above-mentioned RFEH systems …
”
Reviewer#2, Concern # 2: Since the presented work is going to be published, I think it is very helpful for the peer researchers to know the components used. Please give a comprehensive list of the major components used in the RFID sensor and the relay.
Author response: We appreciate the reviewer comment and have added one summary table of major components used in the RFID sensor and the relay (Table 2).
Author action: We added one summary table of major components used in the RFID sensor and the relay:
Table 2. Average power consumption of the smart sensor tag operating.
Component | Component Type | Voltage Supply | Power consumption |
PIC16LF1513 | Microcontroller | 1.8 - 5.5 V | ~90 µW |
M24LR04E-R | IC Tag chip | 1.8 – 5.5 V | ~ 200 µW |
SGP30 | eCO2 sensor | 1.62 – 1.98 V | ~1 mW |
HDC2010 | Temperature and humidity sensor | 1.62 – 3.6 V | ~ 400 nW |
MAX6433 | Voltage supervisor | 0.3 – 6 V | ~1.8 µW |
Total | ~ 1.5 mW |
Reviewer#2, Concern # 3: Please explain the relationship between harvested ac voltage and rectified voltage.
Author response: We appreciate the reviewer for the comment and have added explanation about the relationship between harvested ac voltage and rectified voltage.
Author action: We have added explanation about the relationship between harvested ac voltage and rectified voltage as follows:
“… rectification [33]. The relationship between the harvested AC voltage (Vin) and the rectified voltage (Vrect) is calculated based on [38]:
(12)
where n = 3 is the number of states of the multiplier.
Most components of the …”.
Reviewer 3 Report
This study introduced an enhanced method of improving the efficiency of radio frequency energy harvesting (RFEH) system using relay resonator for food monitoring. The work is interesting. However, the following comments should be addressed before the paper can be considered further:
i) In introduction, the sentence “power is the limiting factor” is not clear and needs to be revised.
ii) The full form of all abbreviations used in the figures should be stated in figure captions.
iii) Some suggestions for future studies should be included.
iv) English language should be improved. I think the authors should use past tense instead of present tense in most sentences. For examples,
(a) In abstract, an enhanced method to improve the efficiency of RFEH system using strongly coupled electromagnetic resonance technology “was” proposed. A relay resonator “was” added… The design of relay resonator “was” based on….
(b) In introduction, …we “applied”…”was” analyzed….
There are many more sentences in the text which should be changed to past tense.
Author Response
Reviewer#3, Concern # 1: In introduction, the sentence “power is the limiting factor” is not clear and needs to be revised.
Author response: We appreciate the reviewer for the comment and have changed the sentence accordingly.
Author action: We have changed the sentence for clarification as: “… color sensor. In these proposed sensor devices, energy which powered the sensor was not the major consideration. Accordingly, these devices …”
Reviewer#3, Concern # 2: The full form of all abbreviations used in the figures should be stated in figure captions.
Author response: We appreciate the reviewer for this comment and have updated the manuscript accordingly.
Author action: We updated the manuscript by adding the full form of all abbreviations:
“Figure 1. (a) The conventional radio frequency energy transfer system without relay resonator and (b) its equivalent circuit model.”
“Figure 2. (a) The enhanced multiplication system with relay resonator and (b) its equivalent circuit model for energy harvesting system.”
“Figure 3. Power efficiency simulation results for the position of relay resonator from the transmitter with respect to the distance between transmitter and receiver resonators.”
Figure 4. The comparison of simulated harvested power efficiency between typical wireless power transfer (WPT) and wireless power transfer using a relay resonator (WPT-R) systems.
“Figure 10. Harvested power comparison of two systems: (a) typical radio frequency energy harvesting system and (b) the proposed radio frequency energy harvesting using relay resonator system.”
Reviewer#3, Concern # 3: Some suggestions for future studies should be included.
Author response: We highly appreciate the reviewer for this comment and have included some suggestions for future studies in the manuscript accordingly.
Author action: We updated the manuscript by adding some suggestions in the “5. Conclusions” part for future studies as follow:
“ .... operational efficiency. For the future work, we aim to design a range-adaptive radio frequency energy harvesting system for further investigating the effect of load variation and position of relay resonator between transmitter and receiver on the system performance.
Funding: …”
Reviewer#3, Concern # 4: English language should be improved. I think the authors should use past tense instead of present tense in most sentences. For examples,
(a) In abstract, an enhanced method to improve the efficiency of RFEH system using strongly coupled electromagnetic resonance technology “was” proposed. A relay resonator “was” added… The design of relay resonator “was” based on….
(b) In introduction, …we “applied”…”was” analyzed….
There are many more sentences in the text which should be changed to past tense.
Author response: We highly appreciate the reviewer for this comment and have modified the manuscript accordingly.
Author action: We have updated the manuscript accordingly.